# Myocarditis and Chronic Inflammatory Cardiomyopathy, from Acute Inflammation to Chronic Inflammatory Damage: An Update on Pathophysiology and Diagnosis

**DOI:** 10.3390/jcm13010150

**Published:** 2023-12-27

**Authors:** Giuseppe Uccello, Giacomo Bonacchi, Valentina Alice Rossi, Giulia Montrasio, Matteo Beltrami

**Affiliations:** 1Division of Cardiology, Alessandro Manzoni Hospital—ASST Lecco, 23900 Lecco, Italy; giuseppe.uccello@yahoo.it; 2Division of Cardiology, Tor Vergata University Hospital, 00133 Rome, Italy; giacomobonacchi1@gmail.com; 3Department of Cardiology, University Hospital of Zurich, 8091 Zurich, Switzerland; valereds@gmail.com; 4Inherited Cardiovascular Diseases Unit, Barts Heart Centre, St. Bartholomew’s Hospital, London EC1A 7BS, UK; giuliamontrasio@hotmail.it; 5Cardiomyopathy Unit, Careggi University Hospital, 50134 Florence, Italy; 6Arrhythmia and Electrophysiology Unit, Careggi University Hospital, 50134 Florence, Italy

**Keywords:** inflammatory cardiomyopathy, myocarditis, sarcoidosis, systemic sclerosis, systemic lupus erythematosus, eosinophilic granulomatosis with polyangiitis, endomyocardial biopsy

## Abstract

Acute myocarditis covers a wide spectrum of clinical presentations, from uncomplicated myocarditis to severe forms complicated by hemodynamic instability and ventricular arrhythmias; however, all these forms are characterized by acute myocardial inflammation. The term “chronic inflammatory cardiomyopathy” describes a persistent/chronic inflammatory condition with a clinical phenotype of dilated and/or hypokinetic cardiomyopathy associated with symptoms of heart failure and increased risk for arrhythmias. A continuum can be identified between these two conditions. The importance of early diagnosis has grown markedly in the contemporary era with various diagnostic tools available. While cardiac magnetic resonance (CMR) is valid for diagnosis and follow-up, endomyocardial biopsy (EMB) should be considered as a first-line diagnostic modality in all unexplained acute cardiomyopathies complicated by hemodynamic instability and ventricular arrhythmias, considering the local expertise. Genetic counseling should be recommended in those cases where a genotype–phenotype association is suspected, as this has significant implications for patients’ and their family members’ prognoses. Recognition of the pathophysiological pathway and clinical “red flags” and an early diagnosis may help us understand mechanisms of progression, tailor long-term preventive and therapeutic strategies for this complex disease, and ultimately improve clinical outcomes.

## 1. Introduction

Acute myocarditis (AM) and chronic inflammatory cardiomyopathy (infl-CMP) share the pathophysiologic feature of myocardial inflammation. A continuum can be identified between these two conditions with persistence of the inflammatory process in infl-CMP beyond the acute phase. Chronic infl-CMP is characterised by persistence of myocardial inflammation, which leads to cardiac dysfunction and remodelling; this can be associated with a wide range of systemic pathologies, from autoimmune to autoinflammatory related disease [1]. Inflammation is a physiological response of the immune system triggered by infectious agents, damaged cells, and toxic substances (Figure 1).

Inflammation may persist over time after an acute presentation or be silent or chronic. Greater effort is required for the recognition of chronic infl-CMP due to its poor outcome in many cases [2]. Cardiac magnetic resonance (CMR) and endomyocardial biopsy (EMB) have an established role in the diagnosis of acute and chronic inflammatory cardiomyopathies, supporting differentiation within this wide spectrum of cardiac diseases [3]. It is necessary to differentiate each of these different diseases more clearly, both clinically and with imaging techniques, and with tissue biopsy when needed, to recognize those that can be treated with targeted evidence-based therapeutic strategies [4]. In this review, we will provide an overview of the clinical manifestation, electrocardiographic, and imaging findings of AM. Furthermore, we will review the most frequent etiologies of chronic infl-CMP and highlight the importance of an integrated approach (clinical presentation + other diagnostics tools) to allow early diagnosis. Specific conditions such as pediatric myocarditis, Kawasaki disease, rheumatic carditis, Chagas disease, myasthenia gravis, thyrotoxicosis, polymyositis, and other rare infectious forms (parasites, protozoa, and fungi) deserve separate discussion and are not addressed in this document.

## 2. Acute Myocarditis

### 2.1. Definition and Diagnosis

AM is defined as an acute inflammation of the myocardium caused by a direct pathogen insult and/or exaggerated immune response [1]. It represents the most typical case of myocardial inflammatory disease, with an incidence of 4–14 per 100,000 individuals globally and a mortality rate between 1% and 7% [5]. Signs and symptoms usually include chest pain, dyspnoea, and palpitations, accompanied by elevation in high-sensitivity troponins as a marker for myocardial damage in the absence of an underlying acute coronary syndrome. Electrocardiographic anomalies often include ST-segment alterations, mimicking an acute coronary syndrome, the presence of ventricular arrhythmias, and/or atrioventricular blocks (AVB). Echocardiographic findings might include new regional wall motion abnormalities and reduced ventricular systolic function [5,6]. Evidence of myocardial edema and/or myocardial scar on CMR, with or without histological evidence of inflammation and necrosis through EMB, is necessary to confirm the diagnosis [6]. The CMR findings for acute myocarditis are defined according to the 2018 Lake Louise criteria, with evidence of myocardial edema by T2-based criterion (increase in T2 relaxation time or increased signal intensity in T2-weighted sequences) in the presence of myocardial fibrosis (increase in T1 relaxation time), late-gadolinium enhancement (LGE), or increased extracellular volume [7]. EMB in routine clinical practice is mostly reserved for those patients presenting with acute heart failure (HF), not responding to medical therapy, or in case of hemodynamic instability due to its invasive nature and reduced availability in low-volume centers [6]. The EMB criteria for myocarditis diagnosis (Dallas criteria) include myocardial immune infiltration, non-ischaemic myocardial necrosis, and immune cells near necrotic myocytes [5,6].

### 2.2. Pathophysiology

One of the better-investigated pathways involved in initiating immune activation during AM is the development of autoimmune reactions targeting the myosin heavy chain 6, a heart-specific protein [8,9]. The subsequent, prolonged activation of Th1 and Th17 cells specifically targeting MYH6 leads to the development of inflammatory cardiomyopathy [10,11]. This has been lately shown in a mouse model of spontaneous autoimmune myocarditis. Interestingly, this study proved that myosin-specific Th17 cells are imprinted in the gut by bacteria of the Bacteroides species. Indeed, mice treated with antibiotics against these bacteria did not progress to end-stage heart disease. Similarly, in humans with acute myocarditis, significantly elevated CD4+ T cell and B cell responses to Bacteroides have been proven [12].

Ongoing immune activation within the heart muscle can result in the development of cardiomyopathy characterized by unfavorable cardiac remodeling, reduced left ventricular (LV) function, and, ultimately, HF. Overall, persistent immune activation within the heart muscle and the continuous inflammation perpetuated by T cells may play a role in the development of cardiomyopathies and HF, thus driving the progression from acute to chronic myocarditis. In support of this hypothesis, the degree of fibrotic alterations serves as a robust predictor of unfavorable outcomes, including higher mortality rates and an increased frequency of hospitalizations among patients with HF. As previously discussed, commensal gut bacteria might promote the progression from acute myocarditis to chronic cardiomyopathy in genetically susceptible individuals [11].

### 2.3. Etiologies

#### 2.3.1. Acute Lymphocytic Myocarditis

Acute lymphocytic myocarditis (ALM) pattern is characterized by an infiltrate of mononuclear immune cells (lymphocytes T, CD3+). The etiology counts many causes: viral infections (with cardiac trophism or molecular mimicry mechanism) and toxic or autoimmune disorders. Common viral pathogens are parvovirus B19, enterovirus, and upper respiratory tract viruses such as coronaviruses, adenovirus, or influenza viruses. ALM is a histological diagnosis from the EMB specimen. EMB is essential to distinguish innocent bystanders from the causative virus of myocarditis. The value of viral presence to guide immunosuppressive treatment in patients with lymphocytic myocarditis remains a subject of debate, especially when considering the different clinical impacts of viruses. However, new interesting data demonstrate that immunosuppression is a possible therapeutic strategy in lymphocytic myocarditis despite the presence of B19V-/HHV6-DNA EMB [12].

#### 2.3.2. COVID-19 and mRNA Vaccine-Related Myocarditis

Myocarditis frequently starts from a viral infection, such as influenza and coronaviruses, amongst others. During the COVID-19 pandemic, a higher incidence rate of myocarditis related to COVID-19 infection, reaching up to 59 to 64 per 100,000 males and 20 to 36 per 100,000 females, was observed [13]. The reasons behind SARS-CoV-19 increased myocardial tropism are still unclear and are likely related to a potentially genetic-driven and virus-induced deranged immune-mediated answer and the widespread endothelial dysfunction associated with microvascular dysfunction in different vascular beds [14]. In correspondence with the COVID-19 outbreak and the development of an appropriate vaccination, a higher-than-expected rate of messenger RNA vaccine-related myocarditis has been observed. Although the underlying mechanisms still remain unclear, both short-term and midterm prognoses have been reported to be favorable, thus prompting vaccination due to a clear benefit compared to associated risks [15].

#### 2.3.3. Giant Cell Myocarditis

Giant cell myocarditis (GCM) represents a separate entity characterized by histological findings of giant cell infiltrates at the level of the myocardium. GCM is a rare disorder presenting often with a fulminant course requiring a prompt histological diagnosis and immediate immunosuppressive treatment [16].

#### 2.3.4. Check-Point Inhibitor Myocarditis

Myocarditis can also be triggered by drugs, such as immune check-point inhibitors (ICIs). Despite being a valuable treatment option for numerous solid cancers, especially for metastatic melanoma, ICI therapy has been linked to systemic immune-related adverse events. Although infrequent, it can lead to myocarditis in approximately 0.3% to 1.2% of cases, ranging from mild to severe clinical pictures, with the latter being associated with a substantial mortality rate ranging from 25% to 50% [17,18]. In the absence of clear imaging criteria, the finding of troponin-elevation during ICI therapy is challenging, and its practical clinical consequences are often unclear [19].

#### 2.3.5. Eosinophilic Myocarditis

This condition is commonly, although not necessarily, associated with laboratory evidence of hypereosinophilia in peripheral blood, and the diagnosis needs to be confirmed with the histological presence of eosinophilic infiltrate on the EMB. More frequently, it occurs in the presence of systemic autoimmune diseases such as Eosinophilic granulomatosis with polyangiitis (EGPA), hypersensitivity reaction to drugs, in the rare hypereosinophilic syndrome with the myeloproliferative or idiopathic form, or in the case of parasitic infections. Eosinophilic myocarditis is a relatively rare condition, and its presentation may vary from paucysintomatic AM to fulminant myocarditis; the inflammation may persist in subacute and chronic forms and evolve into restrictive cardiomyopathy (Loeffler syndrome). Autoptic data show that cardiac damage seems to involve both ventricles with patchy distribution, limiting the EMB sensitivity for the diagnosis. In-hospital death was around 22%, with the highest mortality in the hypersensitivity form, and the most common clinical presentation is HF with severe LV systolic dysfunction. On CMR, the most frequent LGE pattern is extensive subendocardial, extending to mid-wall and subepicardial layers. Intracardiac thrombi are common [20].

## 3. Chronic Inflammatory Cardiomyopathy

Chronic infl-CMPs are a group of different diseases in which myocardial inflammation leads to ventricular dysfunction and dilatation or hypokinetic non-dilated phenotype, generally with a longer duration of symptoms [8]. Persistent inflammation can be secondary to an evolving AM or represent cardiac involvement in the context of a systemic autoimmune disorder or a hypersensitive reaction to toxins and drugs [2]. Although each of these conditions has a different pathophysiology, it is now quite clear that the common element for the development of cardiomyopathy is the immune system activation [21], triggered by different factors. The most frequent clinical presentation, ECG, and imaging findings of systemic inflammatory diseases with cardiac involvement are shown in Table 1

### 3.1. Etiology and Pathophysiology of Chronic infl-CMP Secondary to Acute Myocarditis

As previously described, innate and cellular immunity is activated in viral AM through different pathways that can evolve into long-term cellular damage and myocyte abnormalities. The transition from the acute phase to dilated cardiomyopathy (DCM) is not well defined. Some studies suggest that a mis-activation of M2 macrophages, regulatory cells implicated in immunosuppressive functions, causes a late inflammatory response and progression towards chronic cardiac fibrosis. Moreover, Th-17 lymphocytes and interleukin-17 have been implied in the progression to HF and ventricular dilation [22,23], while in enteroviral infections, some variants of Toll-Like Receptor 3 could blunt innate immune response to the pathogen, leading to a persistence of viral load and increased risk of cardiac disfunction [24]. Some drugs (e.g., dobutamine, beta-lactams, clozapine/olanzapine, smallpox vaccine, carbamazepine, and sulphonamides) implicated in hypersensitivity reactions, parasite infections, and hypereosinophilic syndromes have been associated not only with acute inflammation but also with a progression to DCM, in particular in patients that develop eosinophilia, through a thrombo-fibrotic remodeling process [25]. Autoimmunity plays an important role: in about 30% of patients with myocarditis and idiopathic DCM, it has been established the presence of circulating cardiac autoantibodies, in particular against α- and β-myosin heavy chains, with a controversial pathogenetic and prognostic role [26]. Interestingly, these antibodies have been detected in asymptomatic relatives of patients with DCM—antibodies-positives relatives had a greater risk of progression to LV dysfunction at a younger age—this suggests a role of these molecules in the early development of the disease [27]. In addition, some data propose that evolution to chronic cardiomyopathy can be favored by a genetic predisposition. A locus encoding HLA class I e II proteins has been identified in patients with idiopathic DCM that suggests a link between autoimmune response and evolution to HF [28]. Moreover, Naruse et al. have described the possible role of an HLA allele in cardiac involvement of sarcoidosis [29].

### 3.2. Clinical Features and Diagnosis

Patients affected by inflammatory DCM have signs and symptoms of HF; progressive LV dilation leads to increased filling pressures and volume overload. Evolvement towards DCM occurs at least 30 days after acute myocarditis, but it could be the first presentation of cardiac involvement in patients with systemic inflammatory diseases, with a spectrum of symptoms that range from exertional dyspnoea to interstitial lung changes and pitting edema in more advanced ventricular dysfunction [30]. Markers of cardiac damage, such as troponin and natriuretic peptides, might be only mildly elevated in the chronic quiescent setting or be significantly raised in acute decompensated phases [31]. Electrocardiography shows a variety of nonspecific alterations—ST-T abnormalities, pathological Q-waves, poor R-wave progression in precordial leads, and fragmentation of QRS are common—supraventricular arrhythmias such as atrial fibrillation, sinus pauses, and conduction blocks, are possible. An elevated burden of premature ventricular contractions (PVCs) and non-sustained ventricular tachycardia (NSVT) has been reported [32]. The echocardiography is compatible with DCM features. Progressive LV end-diastolic volume with reduced ejection fraction is associated with signs of diastolic dysfunction and elevated E/e’ ratio; also, a decreased global longitudinal strain in preserved radial LV function can be an indicator of persistent myocardial inflammation [33]. CMR is the most accurate non-invasive technique to assess myocardial fibrosis and chronic inflammation. It is indicated not only in AM but also when persistent cardiac symptoms suggest an evolution towards chronic DCM and can guide EMB in case of “patchy” disease [2]. CMR provides information on differential diagnosis (arrhythmogenic right ventricular cardiomyopathy, idiopathic DCM, sarcoidosis, hypertrophic of infiltrative cardiomyopathy, and ischemic or non-ischemic patterns). Furthermore, the use of LGE allows for the identification of necrosis or replacement fibrosis [34]; its persistence with the disappearance of edema is a marker of unfavorable prognosis, while the mid-wall septal LGE pattern is an independent predictor of cardiac events [35]. Increased extracellular volume, a marker of myocardial fibrosis, has been strongly associated with HF hospitalization with an even greater prognostic value than the ejection fraction [36]. An abnormal pericardial effusion is more common in inflammatory than in idiopathic DCM [37]. Moreover, 18F-fluorodeoxyglucose computed tomography (FDG-PET), through the accumulation of this analog of glucose in metabolically active cells, can be useful to identify myocardial inflammation; its use has been explored particularly in cardiac sarcoidosis, combining analysis of perfusion and inflammation abnormalities and giving information about early and advanced stages of the disease [38]. Lastly, the gold standard for identifying acute and chronic myocardial inflammation and possible etiology remains EBM. In addition, it can provide useful information on viral genomes’ presence by a polymerase chain reaction and add prognostic information through the characterization of infiltrated tissue. A high number of T lymphocytes or macrophages, for example, predicts an increased risk of mortality in the long-term [39].

## 4. Myocardial Involvement in Systemic Inflammatory Diseases

### 4.1. Sarcoidosis

#### 4.1.1. Clinical Presentation

Sarcoidosis is an autoinflammatory disease characterized by non-necrotic granulomatous lesions of unknown etiology and multiple organ involvement [40]. The organs most affected by systemic sarcoidosis (SS) are the lung, hilar lymph nodes, skin, central and peripheral nervous system, liver, and heart. Necropsy studies showed cardiac involvement in 25–85% of cases, demonstrating the presence of granulomas despite a fourth of these patients having been asymptomatic from a cardiac perspective [41,42]. Recent evidence shows a frequency of isolated cardiac sarcoidosis (CS) between 25% and 65%, mostly being diagnosed in the context of cardiac screening in patients with known SS [40,43]. The inflammation can affect any area of the myocardium, including the RV, but mainly affects the basal segments of the interventricular septum, frequently leading to dysfunction of the atrioventricular conduction system and the basal segments of the LV free wall [44]. AVB, ventricular arrhythmias, and HF constitute the triad of cardiac manifestations in CS. The highest prevalence is for II-degree type 2 and III-degree AVB [40]. Ventricular arrhythmias are also common, with non-sustained and sustained ventricular tachycardia originating with a re-entry phenomenon in areas of myocardial inflammation and ventricular fibrillation. Unexpected or aborted sudden cardiac death (SCD) occurs in 14% of patients with CS [45]. Supraventricular arrhythmias, atrial fibrillation, and atrial flutter are also frequent. The most common symptoms are syncope, pre-syncope, and palpitations due to arrhythmias, as well as HF symptoms [46]. Less frequently, CS manifests with unspecific chest pain and pericardial effusion. Signs and symptoms due to the involvement of other organs are essential elements and should prompt clinical suspicion of CS. The most represented extracardiac symptoms are cough and dyspnea due to lung involvement, skin lesions such as erythema nodosum and lupus pernio, and signs and symptoms secondary to central and peripheral nervous system involvement. [47]. It is possible to recognize two courses of SS: a time-limited course, two-thirds of which had self-remitting disease within 1 to 3 years; and a chronic course, in which 10–30% of patients required prolonged treatment [48].

#### 4.1.2. Diagnostics

Routine electrocardiography is useful for cardiac screening of SS patients and has prognostic relevance in predicting cardiac events [49]. It may show a wide range of findings: prolonged PR, fascicular branch block, right or left bundle branch block, pathological Q waves (often with a posterolateral infarction pattern), fragmented QRS complexes, ST segment abnormalities, negative or biphasic T wave, rarely epsilon waves [50]. Among these, conduction abnormalities and fragmented QRS are associated with an increased risk of cardiac events [50]. Moreover, the T peak–T end to QT interval ratio is associated with fatal arrhythmic ventricular events and long-term outcomes [49]. Echocardiography usually does not add more value to clinical presentation and ECG because of its low sensitivity despite a high positive predictive value [41]. Typical echocardiographic findings are the thinning of basal segments of interventricular septum, rarely interventricular septum hypertrophy (due to the presence of edema), and segmental wall motion abnormalities. LV systolic function is affected in the latter stages of the disease [51]. CMR has the highest sensitivity and specificity among imaging techniques [52]. In the early inflammatory phase, typical features are increased intramyocardial intensity of T2-weighted signal, global increase in native T2, T2-STIR, increased wall thickness, and wall motion anomalies with non-coronary distribution [53]. The inflammatory phase is also characterized by LGE due to the presence of edema [41]. The chronic phase, characterized by fibrosis, shows a low T2 weighted signal, increased diffuse LGE signal, and increased extracellular volume with T2 weighted scan [52]. Regarding nuclear imaging techniques, Tl201-SPECT perfusion shows decreased uptake with segmental distribution, while Ga127 scintigraphy presents a higher uptake detecting active inflammation. The latter does not add diagnostic value but can predict the effectiveness of corticosteroid treatment [47]. FDG-PET is another leading advanced imaging modality, useful for the diagnosis and management of CS and for detecting extracardiac disease activity, which can be targeted for tissue biopsy. FDG-PET also has an added value when CMR is negative, but the clinical suspicion is high, and overall, CMR and PET are considered complementary imaging techniques. Recent developments in hybrid CMR and FDG-PET imaging have proven to be promising [54].

### 4.2. Systemic Sclerosis

#### 4.2.1. Clinical Presentation

Systemic sclerosis (SSc) is a multi-organ connective tissue disease characterized by immune system dysfunction, which results in generalized fibrosis. Cardiac injury is bimodal, brought about by vasospasm with ischemia and reperfusion damage, referred to as “cardiac Raynaud’s phenomenon,” and by myocardial inflammation, promoting progressive myocardial fibrosis [55]. According to necropsy studies, cardiac involvement reaches 80% of SSc cases, although it is often occult [56]. The EULAR Scleroderma Trials and Research (EUSTAR) database shows that cardiac causes account for 26% of SSc-related mortality. It has been recognized that 15% of patients with first cutaneous manifestations have a form of SSc with primary cardiac involvement (pCI) [57]. pCI has a worse prognosis. The first clinical manifestations are acute HF (often in the framework of preserved ejection fraction), asymptomatic conduction defects, ventricular arrhythmias, and isolated diastolic dysfunction, which has been defined as the ‘hallmark’ of pCI. [55]. In 90% of cases presenting as multiform PVCs, the most frequent cardiac manifestations are arrhythmias, while sustained ventricular tachycardia and supraventricular arrhythmias are less frequent. SCD is reported in 5% of patients with pCI or musculoskeletal disease [58]. The first clinical presentation can be myocarditis or pericarditis, associated with acute HF symptoms and chest pain. Regardless of the primary myocardial involvement, as the disease progresses, patients develop HF symptoms and asymptomatic and symptomatic AVB. Independently of left ventricular ejection fraction (LVEF), the presence of diastolic dysfunction is associated with higher mortality [59]. Males, older patients, and those with skin ulcers and skeletal muscle involvement have the highest risk of developing myocardial involvement [55]. The most represented autoantibodies in patients with pCI are antitopoisomerase-I [60]. Importantly, the presence of pulmonary hypertension, often secondary to lung involvement in SSc, must be carefully ruled out and followed up over time [58].

#### 4.2.2. Diagnostics

The electrocardiogram was abnormal in 28% of patients in large registries [61]. The most common ECG changes were multiform PVCs, pseudo-infarct Q waves, atrioventricular conduction abnormalities, and left and right bundle branch blocks [61]. Similarly, Holter ECG exams showed high ventricular arrhythmias burden (PVCs, ventricular tachycardia) and supraventricular arrhythmias [55]. Signs of autonomic dysfunction can be recognized using Holter ECG analysis in the form of a reduced heart rate variability and through exercise testing as an increase in heart rate time recovery [62]. Speckle tracking echocardiography (STE) allows early detection of diastolic and systolic dysfunction before the onset of HF symptoms [63]. In a large registry study, patients with low LVEF were 5.5%, whereas global longitudinal strain and global circumferential strain proved to be effective in demonstrating deterioration of contractile function despite normal LVEF in patients with diastolic dysfunction [64]. In older studies, SPECT could demonstrate rest- or stress-induced myocardial perfusion defects in the absence of coronary artery disease [65]. Fixed perfusion defects are related to the presence of a scar. CMR showed systolic and diastolic dysfunction in 20% of SSc patients [66]. Typical findings are the presence of fibrosis detected with LGE in a patchy, non-ischemic (often mid-wall) distribution, even at the early stages of the disease in the absence of clinical cardiac manifestations [66]. The amount of LGE correlates with ventricular arrhythmic burden [67]. In a recent study, T1 mapping showed an increase in extracellular volume in 50% of patients with SSc. One-third of these had had negative conventional cardiology screening tests [68]. A significant rate (11%) of silent myocarditis has been demonstrated in SSc in the absence of laboratory or clinical indices of cardiac involvement. However, the presence of edema in T2 mapping sequences is much less frequent in asymptomatic patients [67].

### 4.3. Systemic Lupus Erythematosus

#### 4.3.1. Clinical Presentation

Systemic lupus erythematosus (SLE) is a chronic autoimmune rheumatic disease with frequent heterogeneous multi-organ involvement and numerous clinical phenotypes. SLE etiology consists of a complex interaction of genetic, epigenetic, and environmental factors, with auto-antibodies as a serological hallmark, with more than 95% of cases presenting ANA autoantibodies [69]. Risk factors for cardiac involvement are male gender, at least one point in the Systemic Lupus International Collaborating Clinics/American College of Rheumatology damage index score, and the presence of anti-Sm or anti-Ro antibodies [70]. The main cardiac manifestation is myocarditis, in isolation or associated with pericarditis. Myocarditis frequency has decreased with the advent of immunosuppressive therapy, prior to which myocarditis was observed in 40–70% of SLE patients on autoptic studies [70,71]. With current state-of-the-art therapy, myocarditis in SLE ranges between 15% and 25% of patients or even less on autoptic studies [72]. The frequency of SLE-related myocarditis varies between different ethnicities, being higher in patients of African and Asian descent [73]. Cardiotoxicity, particularly secondary to chloroquine/hydroxychloroquine, needs to be ruled out in suspected SLE-related myocarditis cases in patients treated with these drugs [71]. Among rheumatic diseases involving the heart, SLE is the one mostly associated with cardiogenic shock, but it can also be asymptomatic [74]. Symptomatic pericarditis occurs in around 25% of SLE patients, but asymptomatic forms are more frequent [75]. Tamponade can be the first manifestation [76]. The presence of pericarditis alone is infrequent. Coronary artery disease has a high incidence in SLE due to vasculitis and accelerated atherosclerosis. Large registry studies determined a >10-fold higher risk of coronary artery disease in young SLE patients and an absolute risk ranging from 5% to 12% in patients < 70 years of age [77]. Ischemic cardiac disease has been described as a leading cause of SLE patients’ mortality [78]. Valvular abnormalities are also common: in a recent meta-analysis, 23.3% of SLE patients presented with valvular abnormalities, consisting mostly of mitral and tricuspid regurgitation, with frequent findings of vegetations or leaflet thickening [79]. Pulmonary arterial hypertension (PAH) is found in 4–9% of SLE patients, and it is often caused by vascular remodeling and associated with higher mortality due to right-sided HF [80].

#### 4.3.2. Diagnostics

Electrocardiography can be useful in patients with symptoms and signs of pericarditis or myocardial ischemia. AVB and interventricular conduction abnormalities are less frequent compared to other systemic immune-mediated diseases, although they are more common in the pediatric population due to high serum levels of anti-Ro antibodies more pronounced in neonatal SLE [81]. Although past studies have emphasized the presence of non-infectious vegetation in SLE patients, current registries show a very low frequency of Libman Sacks endocarditis [79]. Another recent study shows how speckle tracking echocardiography can predict quite early in the course of the disease the presence of systolic dysfunction, independent of disease activity calculated with the SLE disease activity index [82]. Reduced global longitudinal strain has also shown a prognostic value [83]. Furthermore, due to vasculitis-related coronary artery disease, it is not uncommon to detect regional wall motion abnormalities or aneurismal wall thinning. An accurate study on differences between the CMR findings in acute infective myocarditis and active SLE shows both a high T2 intensity signal and a high gadolinium enhancement intensity signal [84]. While LGE was more pronounced in infective myocarditis, it was present in a minority of patients with SLE-related myocarditis. Only 20% of patients with active SLE fulfilled clinical criteria for diagnosis of active myocarditis, while CMR was positive for myocarditis in 80% of patients [58].

### 4.4. Eosinophilic Granulomatosis with Polyangiitis

#### 4.4.1. Clinical Presentation

Eosinophilic granulomatosis with polyangiitis (EGPA) is a rare (05–4.2 per million) necrotizing granulomatous vasculitis with multi-organ involvement, originally described with late-onset asthma and chronic sinusitis, small-to-medium vasculitis and peripheral blood and tissue eosinophilia and currently diagnosed with clinical criteria (nasal and cartilaginous involvement and conductive or sensorineural hearing loss) plus laboratory, imaging and biopsy criteria [85]. It is considered an ANCA-associated vasculitis; however, ANCA antibodies are present only in 40% of these patients. Histopathological findings of ANCA + EGPA are small vessel vasculitis, while in ANCA-EGPA, eosinophilic tissue infiltration is more prevalent. Risk factors for myocardial involvement are high eosinophilic peripheral blood count and absence of ANCA antibodies. Cardiac involvement is present in a quarter of patients with EGPA, associated with high peak eosinophilic count. Persistence of eosinophilic tissue infiltration results in the progression of fibrosis, which can evolve in either dilated/non-dilated cardiomyopathy with impaired LV systolic function or in a restrictive phenotype, typically called Loeffler syndrome [25,86]. The latter patients have poor prognosis, particularly if untreated. Fulminant myocarditis is a possible manifestation of myocardial involvement, and an in-hospital mortality of 21.7% has been reported [25]. The spectrum of cardiac involvement is broad, and acute myocarditis with regional wall motion abnormalities, biventricular involvement, and apical thrombi and aneurysms are common manifestations. Intracavitary thrombi are a common finding in both ventricular chambers [87]. Angina is a frequent presentation, as EGPA is a vasculitis of the small and medium vessels, and microvascular dysfunction is common [58].

#### 4.4.2. Diagnostics

The ECG often shows intraventricular conduction disturbances; the right bundle branch block and left posterior fascicular block appear to be more common because of more frequent right ventricular involvement, while AV blocks are a quite rare manifestation. ST-segment and T-wave abnormalities are frequent due to vasculitis. Echocardiographic assessment (with contrast if needed) is the first step to identify wall motion abnormalities, pericarditis, apical aneurysm, and intracardiac thrombi. On CMR, the most prevalent, typically LGE, is seen in an extensive subendocardial pattern with non-coronary distribution, occasionally with transmural distribution. Also, in patients with clinical remission, cardiac involvement with subclinical inflammation together with fibrosis is a common finding, demonstrated using early gadolinium enhancement and T2-weighted imaging [88]. In patients with hypereosinophilia and/or EGPA who are asymptomatic from a cardiac perspective, identification of cardiac involvement is based on ECG and echocardiography, while advanced imaging should be reserved for selected patients or in case of acute symptomatic presentations.

### 4.5. Psoriasis

#### 4.5.1. Clinical Presentation

Myocardial inflammation in the context of psoriasis (PS) is a rare but described finding. The pathogenesis of PS is still not entirely clear, and it is probably secondary to a complex interaction between environmental factors, genetic background, and immune system activation. The largest study to date is from Eliakim Raz et al., who described a case series of 2292 patients hospitalized for PS. These patients had an incidence of cardiomyopathy of only 1% and DCM between 0.5 and 0.8% [89]. In a more recent case series on 100 patients with moderate to severe forms of PS, 5% had a DCM detected. All patients showed lymphocytic myocarditis and presented anti-heart antibodies cross-reactive to the skeletal muscle. Furthermore, a 2.48-fold overexpression of IL-17A was associated with myocarditis in these patients [90]. These cited studies and case reports showed an acute onset with severely depressed systolic function as the typical form of myocardial involvement in PS. Interestingly, a large real-world study on about 2 million patients demonstrated that PS is not an independent risk factor for DCM after normalization for cardiovascular risk factors, but it is independently associated with viral myocarditis with an OR of 2.3 [91].

#### 4.5.2. Diagnostics

Pericardial effusion is a finding frequently reported during echocardiographic examination in PS patients. Biventricular dysfunction with severely reduced ejection fraction is present in half of the cases when the heart is involved. Conduction disorders are reported sporadically, while ventricular and supraventricular arrhythmias are common. CMR showed high native T1 and T2 values associated with the hyperintensity of STIR T2 weighted images, demonstrating edema [90].

## 5. Arrhythmogenic Right Ventricular Cardiomyopathy with “Hot Phase” Episodes

The morphological spectrum of arrhythmogenic right ventricular cardiomyopathy (ARVC) has broadened in the past years and has come to include variants with predominant or even isolated LV involvement. A recent paper from Corrado et al. suggests a new definition and classification of arrhythmogenic cardiomyopathy, including the disease that involves the left ventricle. New recognition of the diagnostic role of CMR provides unique information on myocardial tissue characterization for the detection of myocardial scar to characterize biventricular and left disease variants and exclude other “non-scarring” myocardial diseases [92]. To confirm this new clinical finding, a new spatial transcriptomics study highlights changes in gene expression to identify novel drivers of cardiomyocyte loss for specific remodeling processes occurring during ACM [93]. Myocarditis and recurrent myocarditis are distinct and often under-recognized presenting phenotypes of ARVC, which reflect an active or so-called “hot-phase” of the disease. Approximately 50% of probands are found to carry a pathogenic/likely pathogenic (P/LP) variant in one of the desmosomal genes, which is usually inherited with an autosomal-dominant pattern [94]; however, Brodhel et al. demonstrated that a homozygous deletion of desmocollin-2 encodes a truncated form of the protein with an autosomal-recessive transmission [95], while a desmin de-novo mutation has been founded in a 17-year-old patient with severe disease and with no relatives affected [96]; this evidence suggests a genetic analysis in ARVC patients without a clear genetic history should always be considered. The differential diagnosis between these two entities (isolated AM versus ARVC) can be challenging; however, it remains crucial, and genetic testing and family screening play a major role.

### 5.1. Pathophysiology

In ARVC, mutations in desmosomal genes compromise intercellular adhesion, ultimately leading to myocyte necrosis and fibro-fatty replacement of myocardial tissue, which promotes electrical instability, ventricular arrhythmias, and SCD, but also progression towards right ventricular/LV dilatation, dysfunction and HF. Disease progression is considered to occur in bouts rather than as a continuous process. These periodic exacerbations of an otherwise-quiescent disease are clinically characterized by chest pain and myocardial enzyme release. An underlying inflammatory/immune mechanism to ARVC has been reported since 1996 when focal lymphocyte infiltrates and myocyte necrosis consistent with myocarditis were described in up to 67% of ARVC hearts on post-mortem examination [97]. Inflammatory infiltrates correlate with more severe structural cardiac abnormalities, implicating inflammatory triggers as a potential modulator of ARVC deterioration. To date, it remains to be clarified whether an inflammatory trigger is the “primum movens” that determines cardiomyocytes’ damage, necrosis, and the consequent repair process with fibrofatty infiltration or if inflammation is rather a reactive phenomenon following myocyte loss [98]. It has been suggested that genetically determined desmosomes malfunction predisposes cardiomyocytes to detachment and death and renders them more susceptible to injury by infectious pathogens [99]. Cardiotropic viruses have been detected in the EMB of ARVC patients. They had been initially postulated to contribute to the onset and progression of the disease; however, it seems more likely that viruses are benign bystanders rather than actual causative triggers. Another known trigger for acute inflammatory flares in ARVC is physical exercise [100]. Finally, an additional autoimmune response against intercalated disk components and myosin has been recently identified, hence fostering the idea of a role for autoimmunity in pathogenesis and disease progression [101].

### 5.2. Clinical Findings and Genetics

In a large cohort of ARVC patients, those presenting with myocarditis-like episodes are usually young (mean age 26 years ± 14 years), and hot phases are often the first presentation in the pediatric population [102]. The clinical presentation resembles that of an AM, with chest pain and enzyme release in the context of unobstructed coronaries. The anamnesis can be helpful, as ARVC patients more often exhibit a personal history of previous or recurrent myocarditis, whereas prodromal flu-like or viral-illness-related symptoms are infrequent when compared to classical acute myocarditis [103]. Chest pain is often accompanied by transient ischemic ST-segment changes on 12-lead ECG on a background of abnormal QRS voltages with fractionation and negative T waves in the right ventricular or lateral leads, although up to a third of patients have a normal ECG at presentation. Hemodynamic instability at presentation is rare, but episodes of sustained ventricular arrhythmias and SCD at presentation have been described [104]. NSVT and increased burden PVCs can be commonly seen on in-hospital telemetry monitoring. Troponin elevation is almost invariably present, whereas notably, C-reactive protein and leukocytes are usually not elevated during the hot-phase [98]. ARVC should also be suspected in those individuals presenting with myocarditis who have a family history of recurrent myocarditis, SCD, or cardiomyopathy. Genetic testing in this subset of myocarditis patients is an essential tool supporting the diagnosis of ARVC, and it should always be considered in case of a positive family history of cardiomyopathy or SCD [105]. Desmoplakin (DSP) is the most common gene involved, accounting for 69% of cases, followed by plakophilin-2 (PKP-2), desmoglein-2 (DSG-2) desmin (DES), and the junction plakoglobin (JUP) gene, associated with Naxos disease; the mutation of other genes beyond desmosome has also been demonstrated to a lesser extent, like phospholamban (PLN), lamin A/C (LMNA), integrin-like kinase (ILK), titin (TTN) or genes encoding for area composita like cadherin-C (CDH-2) and alpha-T-catenin (CTNNA3). The DES mutation causes a predominant LV phenotype of arrhythmogenic cardiomyopathy with a high incidence of adverse clinical events. The pathophysiologic mechanism seems to be correlated to an abnormal desmin dimer and oligomer assembly and its connection with membrane proteins [106]. Moreover, new findings provide evidence that ILK is a new cardiomyopathy disease gene associated with arrhythmogenic phenotype in two unrelated families; in particular, two missense variants (p.H33N and p.H77Y) seem to be associated with structural damage of ILK [107]. PLN regulates the sarcoplasmic reticulum Ca^2+^ pump, maintaining regular Ca^2+^ homeostasis. Mutations in genes encoding PLN are related to both phenotype of dilated cardiomyopathy and arrhythmogenic cardiomyopathy with a strong relationship with sudden cardiac death at a young age and ventricular arrhythmic events [108]. Patients with DSP-cardiomyopathy have a greater history of acute myocardial injury [98], while in a murine model, the overexpression of desmocollin-2 (DSC2) has been associated with myocardial inflammation and fibrotic remodeling [109]. The role of these “hot-phases” in disease progression and arrhythmic risk stratification remains unclear. Bariani et al. did not find a difference in terms of adverse outcomes in ARVC patients experiencing myocarditis episodes compared to those who did not experience a ‘hot phase’ [98], although in few patients, inflammatory episodes preceded a worsening of LV systolic function or an episode of ventricular tachycardia [110]. Particularly, truncating DSP mutations have been consistently associated with more aggressive phenotypes and higher occurrence of ventricular arrhythmias and SCD in the absence of severe systolic dysfunction [103]. Finally, a minority of ARVC patients presenting with acute inflammation do not demonstrate a pathogenic variant in desmosomal genes and are considered to have a “gene-elusive” form of inherited ARVC (based on clinical presentation and imaging tests). In these patients, recurrent hot phases, together with the evidence of arrhythmias and the extent of scar and systolic dysfunction on CMR, should be considered for decisions related to primary prevention of implantable cardioverter defibrillator implantation [103].

### 5.3. Diagnostics

CMR plays a central role in the diagnosis of acute inflammation and ACM. It should always be performed, whenever possible during the acute phase. Fatty replacement on CMR, seen as regions with chemical shift artifacts on cinematic sequences and elevated T2 values correlating with disease activity, may help identify such ARVC patients with acute inflammatory presentation [100]. While the detection of LGE with spotty distribution in the ventricular wall is highly suspicious for ARVC, a sub-epicardial distribution in the appropriate clinical context should raise the suspicion of an underlying inherited ARVC [111]. Typically, these patients present with only mild LV impairment and no or only mild concomitant wall motion abnormalities despite extensive sub-epicardial LGE, which in the most severe cases is circumferential (“ring-like”), resembling the band of fibrous tissue described by pathologists in ARVC hearts [103]. This pattern is peculiar for DSP cardiomyopathy, although it has also been observed in gene-elusive cases, in which the extent of LGE has a strong correlation with arrhythmic adverse events [112]. Finally, patients with hot-phase ARVC tend to have a higher burden of LGE+ segments on follow-up CMR than non-inherited AM [103] (Figure 2, Figure 3 and Figure 4).

FDG-PET detects inflammation and generally is non-specific for the differential diagnosis between ARVC and AM and is not routinely performed. A positive FDG-PET can be found in almost half of ARVC patients in different phases of the disease [113]. However, although FDG-PET can be useful in the differential diagnosis of sarcoidosis due to extracardiac uptake often seen in the latter, these two entities cannot always be distinguished using this method. In a cohort of patients with positive PET undergoing cardiac transplantation for sarcoidosis, histological examination of the explanted hearts revealed that three out of eight patients were actually suffering from ARVC [114]. A positive FDG-PET in a patient with persistent troponin elevation should trigger further diagnostic steps, such as an EMB, especially in the presence of arrhythmias [113].

## 6. Endomyocardial Biopsy: When and Why

EMB remains the gold-standard technique to diagnose AM. It provides relevant prognostic information and can guide treatment. EMB is recommended in patients with complicated AM, with the presence of LV or biventricular systolic dysfunction, severe conduction abnormalities, and ventricular arrhythmias [115]. The American Heart Association recommends EMB as a first-line diagnostic tool in all unexplained acute cardiomyopathy complicated by symptomatic ventricular tachycardia, severe brady-arrhythmias, or hemodynamic instability requiring vasopressor/inotropes or mechanical circulatory support [116]. However, EMB is still underused due to a lack of experience in non-tertiary centers and variable diagnostic performance. Important factors to be considered include clinical pre-test probability, timing and site of sampling (RV vs. LV), and type of myocarditis [117]. The EMB results need to be interpreted in the clinical context and integrated with clinical, laboratory, and imaging findings to achieve higher diagnostic accuracy. In patients with acute HF without hemodynamic instability or arrhythmias and suspicion of inflammatory substrate, an EMB may be considered after CMR to define the type of inflammatory infiltrate before starting an immunosuppressive treatment [115,118]. Histologically, infl-CMP is distinguished by focal or diffuse fibrosis with classic inflammatory infiltrates. Recent studies demonstrate interesting results using electroanatomic mapping-guided EMB and fluoroscopic-guided EMB, improving the diagnostic yield of the procedure [118]. EMB may be particularly useful in case of suspicion of sarcoidosis, GCM, or EGPA. At present, EMB has a low sensitivity in cardiac sarcoidosis due to the often patchy distribution of granulomas [41]. However, the presence of monocyte infiltration and at least moderate fibrosis is a major criterion for diagnosis of sarcoidosis in the 2016 Japanese Cardiovascular Society on diagnosis and treatment of Cardiac Sarcoidosis [119], whereas the 2014 Heart Rhythm Society expert consensus considers as a histological criterion only the recognition of non-caseous granulomas [120]. The characteristic findings on EMB in eosinophilic myocarditis in EGPA patients are eosinophilic infiltrates, necrotizing small and medium vessel vasculitis, as well as fibrinoid necrosis. EMB is often associated with nonspecific findings, such as myocardial fibrosis [121]. In EGPA, the right ventricle is also often affected: this makes the use of RV-EMB useful, particularly in ANCA-negative cases with more frequent and earlier cardiac involvement [88]. In SLE patients with cardiac involvement recognized using CMR, EMB shows low sensibility, inferior to 50%, even with immunohistological techniques [84]. EMB in CSS patients commonly shows the presence of slightly inflammatory infiltrates and extensive fibrosis [122]. Finally, there is no pathognomonic finding in SSc EMB, but the presence of band necrosis, due to reperfusion injury caused by ‘cardiac Raynaud’s phenomenon’, is common [55].

Figure 5 proposes a practical algorithm for early diagnosis in Infl-CMP. Close collaboration between general practitioners, cardiologists, and other specialties (such as immunologists and rheumatologists) is key to reaching early diagnosis.

## 7. Management

Infl-CMPs are a broad spectrum of conditions for which appropriate management depends on the mode of presentation and, most importantly, on the underlying disease. In viral acute myocarditis, there are no targeted therapies [6] other than supportive measures, although the use of anti-viral drugs should be considered in herpes virus infection [123]. In these cases, treatment of complications, such as arrhythmias and HF, and the restriction of physical activity for 6 months are the most important measures [6]. When LV systolic dysfunction occurs, guidelines-directed HF medical therapy is indicated, while in acute HF or cardiogenic shock, such as in fulminant myocarditis cases, hemodynamic and mechanical circulatory support should be considered early [124].

In myocarditis secondary to autoimmune or other systemic disorders, treatment of the underlying condition is key. The use of corticosteroids is often the first line of treatment, frequently in combination with intravenous immunoglobulin, cyclophosphamide, or rituximab during the acute phase, while for maintenance therapy, mycophenolate mofetil, azathioprine, or methotrexate are generally preferred. Guidelines and RCTs on such treatments are scarce. In a controlled trial in patients with new-onset DCM, 16% of which had biopsy-proven myocardial inflammation, high-dose intravenous immunoglobulins did not improve LVEF [125]. Moreover, the first study to assess the efficacy of immunosuppression in AM demonstrated that prednisone combined with azathioprine or ciclosporin did not have a positive effect on LVEF or survival in patients with AM [126]. Subsequent studies have clarified the efficacy of immunosuppression in terms of cardiac function but not in terms of survival benefit [127] and therapy often remains eminence- rather than evidence-based. A long-term benefit of immunosuppression on NYHA class and ventricular function has been seen in patients with DCM and HLA upregulation on biopsy specimens [128]. Some data also suggest remission and HF recovery in giant cells and ICI-related myocarditis [129]. In the TIMIC trial, patients with HF and chronic infl-CMP with no evidence of myocardial viral genomes were randomized to prednisone plus azathioprine for six months versus placebo; the treatment group experienced a significant decrease in LV volumes and LVEF improvement [130]. Finally, a promising approach may be the use of specific monoclonal antibodies directed against molecules involved in autoimmunity: in a case series reported by Frustaci et al., the addition of secukinumab, an interleukin-17A inhibitor, to standard immunosuppressive therapy was associated with complete ventricular function recovery in patients with psoriasis-relate myocarditis [90]. Further studies are needed to confirm this preliminary data and extend it to other autoimmune diseases.

## 8. Conclusions

Infl-CMPs are a group of diseases characterized by myocardial inflammation and often evolve towards LV dilatation and systolic dysfunction. Progression from an acute phase to a chronic one is not always predictable and often depends on the underlying etiology and mechanism of myocardial damage. Nowadays, CMR and EBM are the diagnostic techniques that allow a more accurate characterization, while genetic testing is useful in those cases where an underlying inherited cardiomyopathy is suspected, with significant prognostic implications for patients and their family members. The efficacy of immunosuppression has been demonstrated only in specific diseases, with controversial benefits on survival. Further studies focusing on both pathophysiology and diagnostics are needed to provide further understanding of the disease. Close collaboration between cardiologists and other specialists should be promoted to identify cardiac involvement promptly and define therapeutic strategies and prognostication.

## Figures and Tables

**Figure 1 jcm-13-00150-f001:**
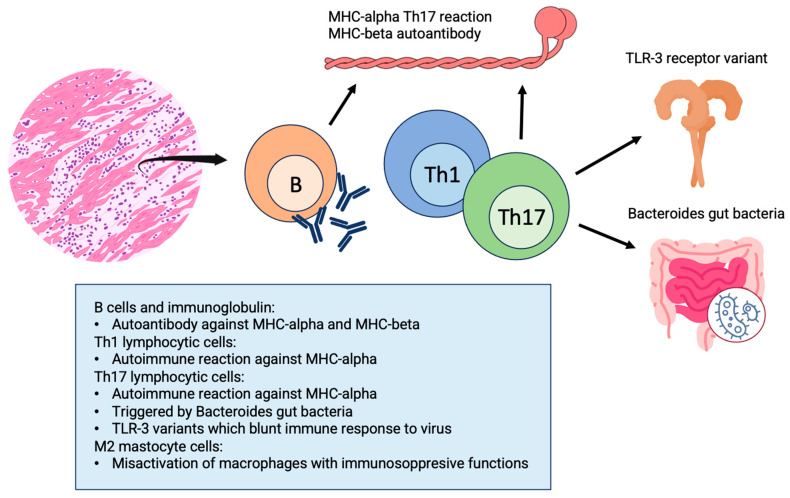
Physiopathological processes leading to the persistence of inflammation and the progression from acute myocarditis to chronic inflammatory cardiomyopathy: TLR-3 variants cause reduction in efficacy of innate immune system response against viruses; Bacteroides species triggered autoimmune reaction against cardiac epitopes; antibodies against alpha- and beta-myosin heavy chain constitute a component of the autoimmune response against the heart. MHC, myosin heavy chain; TLR-3, Toll-like receptor-3.

**Figure 2 jcm-13-00150-f002:**
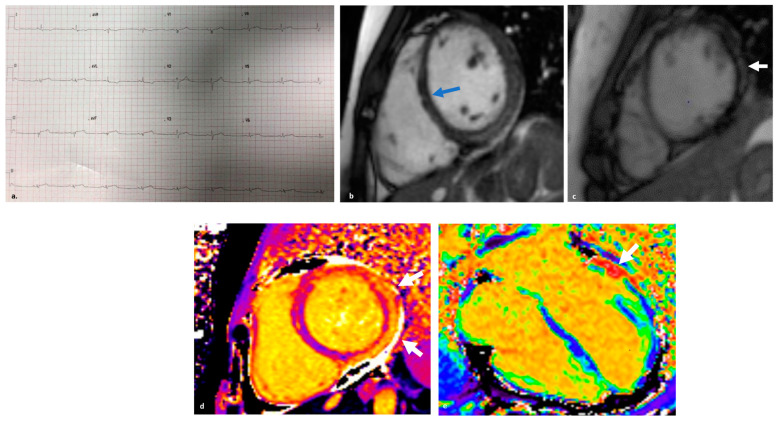
(**a**) ECG during a hot-phase of arrhythmogenic right ventricular cardiomyopathy, showing low-voltages, QRS fractionation as well as ST-elevation in the inferior leads and T waves inversion with borderline ST-segment depression in leads I-aVL. (**b**) Cardiac magnetic resonance (CMR) scan is performed during the acute/hot-phase. She was later found to be heterozygous for a pathogenic nonsense variant in Desmoplakin (DSP) gene: c.478C > T p.(Arg160Ter). Cine steady-state free-precession images showing a non-dilated left ventricle (LV) with increased wall thickness in the lateral wall, in keeping with myocardial edema. Intrinsic contrast on cine-imaging in the areas of edema and scar. There is a small patch of fat infiltration in the mid-septum (blue arrow). (**c**) Irregular LV lateral wall in the lateral segment, which likely represents epicardial fat, infiltrating the LV wall (white arrow). (**d**) T1 mapping is significantly elevated in the inferolateral wall (1790 ms to 1565 ms; normal range 970–1050 ms). (**e**) Extracellular volume is elevated in a patchy pattern (79% in basal anterolateral wall).

**Figure 3 jcm-13-00150-f003:**
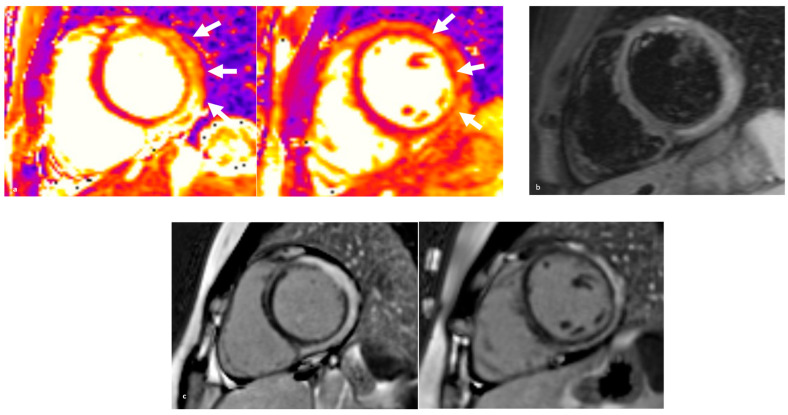
(**a**) T2 mappings are massively elevated in the lateral wall (white arrows) (max 119 ms in basal inferolateral, 85 ms in basal anterolateral, normal range 40–54 ms) and within normal range elsewhere. (**b**) Increased myocardial signal at mid anterolateral and mid inferolateral wall on T2 STIR sequences. (**c**) ”Ring-like” subepicardial late gadolinium enhancement (LGE), becoming almost transmural towards the anterolateral LV wall.

**Figure 4 jcm-13-00150-f004:**
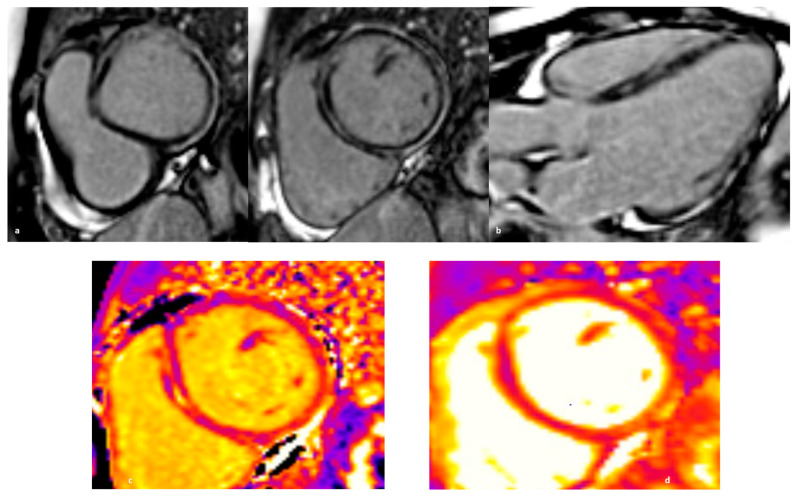
CMR scan for the same patient performed 3 months after resolution of the “hot phase.” The initially edematous and thicker lateral wall has evolved towards thinning and hypokinesia. Compared to the previous CMR scan, the lateral segments appear less edematous (although T2 values remain mildly elevated in some regions) and have also thinned. The extent of scar has not significantly changed. (**a**) Extensive ‘ring-like’/circumferential basal to apical mid-wall LGE, becoming epicardial at the basal to mid anterior, anterolateral, inferolateral, and inferior wall segments. (**b**) Three chamber long axis view of LGE sequences—the amount of scar has not changed, but a thinning of the LV walls can be appreciated. (**c**) Elevated native myocardial T1 value in patches, with an improvement compared to 3 months before, but representing persistent edema (1240–1359 ms). (**d**) Mildly elevated native myocardial T2 values (47–59 ms) in patches improved compared to the previous scan.

**Figure 5 jcm-13-00150-f005:**
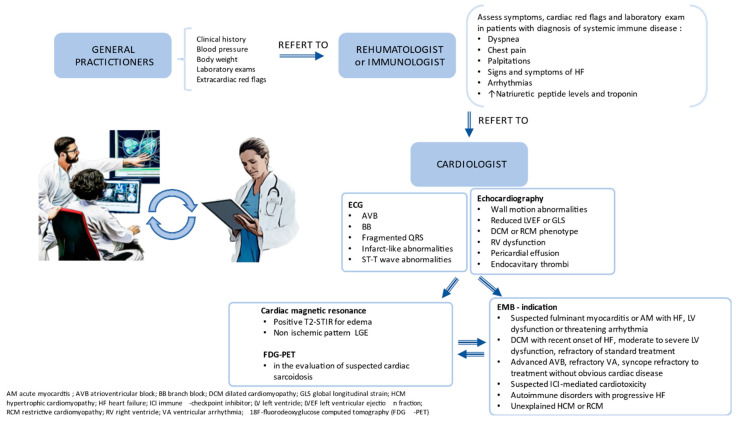
A proposed practical approach algorithm to facilitate early diagnosis of Infl-CMP, with a close collaboration between general practitioners, cardiologists, and other specialties.

**Table 1 jcm-13-00150-t001:** The most frequent clinical presentation, ECG, and imaging findings of systemic inflammatory diseases with cardiac involvement.

Disease	Clinical Presentation	ECG	Echocardiography	CMR	Characteristic Systemic Manifestation for Differential Diagnosis
Sarcoidosis	Advanced AVB VA and SCD HF SVA (AF, AFl) Pericardial effusion	PR elongation Branch block and fascicular block PVCs Fragmented QRS Infero-lateral infarct pattern ST segment and T wave abnormalities	Non-ischemic wall motion abnormality IVS thinning (chronic phase) or hypertrophy (acute phase) LV and RV systolic dysfunction	Subendocardial, mid-wall, subepicardial LGE Patchy distribution	Pulmonary involvement Uveitis Skin lesions (Lupus Pernio, erythema nodosum) Central and peripheral nervous system involvement
Systemic sclerosis	VA Diastolic dysfunction AM HF SVA Pulmonary Hypertension	Multiform PVCs Branch block AVB Pseudo-infarct Q	Diastolic dysfunction Pericardial effusion LV and RV systolic dysfunction Pulmonary Hypertension	Subepicardial and IVS mid-wall LGE	Skin ulcers and diffuse fibrosis Raynaud phenomenon Pulmonary fibrosis Scleroderma renal crisis
Systemic lupus erythematosus	Pericarditis AM, fulminant myocarditis HF CS (vasculitis, accelerated atherosclerosis) Hypertensive cardiomyopathy Valvular heart disease	PR depression ST segment and T wave abnormalities	Pericardial effusion LV Systolic dysfunction with or without ischemic wall motion abnormalities Libman-Sacks endocarditis	Subendocardial, mid-wall, subepicardial LGE	Photosensitive malar rash Arthritis and myositis Glomerulonephritis Gastroenterological involvement Neuropsychiatric lupus
Eosinophilic granulomatosis polyangioiitis	Eosinophilic myocarditis, fulminant myocarditis HF RCM or DCM Pericarditis, pericardial tamponade or constriction Microvascular angina	ST segment and T wave abnormalities	LV systolic disfunction RCM or DCM phenotype Peff Ventricular thrombus	Subendocardial LGE Patchy distribution	Asthma or chronic sinusitis Peripheral blood eosinophilia Gastroenterological involvement Skin hemorrhagic lesions (Palpable purpura) Peripheral neuropathy

Acute myocarditis; AF atrial fibrillation; Afl atrial flutter; AVB atrioventricular block; CS coronary syndrome; DCM dilated cardiomyopathy HF heart failure; IVS interventricular septum; LV left ventricle; RCM restrictive cardiomyopathy; RV right ventricle; SVA supraventricular arrhythmia; VA ventricular arrhythmia; SCD sudden cardiac death.

## Data Availability

Not applicable.

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
