# Peer review of "Myocarditis and Chronic Inflammatory Cardiomyopathy, from Acute Inflammation to Chronic Inflammatory Damage: An Update on Pathophysiology and Diagnosis"

_jcm, 2023, doi:10.3390/jcm13010150_

Round 1

Reviewer 1 Report

Comments and Suggestions for Authors

Thank-you for considering my comment

Author Response

We are grateful to the reviewer for the words of appreciation

Reviewer 2 Report

Comments and Suggestions for Authors

This is a well written paper. Your Title is very broad. You have not commented about your exclusion of rheumatic fever and Kawasaki disease which is quite reasonable though both conditions may be considered under the title “myocarditis”. It would be reasonable for you to state that upfront and give the reasons why, accepting that although you are discussing inflammatory processes involving the myocardium, generally rheumatic fever and Kawasaki disease are considered separately and that your paper will not be further commenting about them.

There are multiple issues with respect to the use of the English language, for example see line 17, 22, 270, 338, 356, 361, 362, 372, 375, etc. It would be most important that the paper is reviewed carefully by a native English speaker prior to resubmitting the paper to the Journal. You may also need to comment as to whether the figures that you have provided are ones that you have created yourself or whether they come from previous articles. If the latter it would be important that you obtain copyright permission for them to be published in another Journal.

Comments on the Quality of English Language

See comments to the author

Author Response

Reviewer Comments:

This is a well written paper. Your Title is very broad. You have not commented about your exclusion of rheumatic fever and Kawasaki disease which is quite reasonable though both conditions may be considered under the title “myocarditis”. It would be reasonable for you to state that upfront and give the reasons why, accepting that although you are discussing inflammatory processes involving the myocardium, generally rheumatic fever and Kawasaki disease are considered separately and that your paper will not be further commenting about them.

We are grateful to the reviewer for the words of appreciation and the constructive comments.

Thanks for your suggestion, pint well taken. Myocarditis and chronic inflammatory cardiomyopathies are a large spectrum of diseases. Rheumatic disease and Kawasaki disease are a complex and underestimated topic, in the first case due to an altered perception in clinical practice that identifies it only as a valvular pathology, in the second due to the greater inflammatory response particularly in medium and small vessels with myocardial ischaemia due to vasculitis. Although these two pathologies also agree in having an inflammatory myocardial involvement associated with the main manifestations previously cited, we have not been able to treat them, as other rheumatological and immunological conditions that have been previously considered, before the work was written. This is because we had to consider limited writing space and wanted to make the text as concise as possible in order to treat the most frequent topics introduced in the most accomplished manner. In particular Kawasaki disease continues to be a diagnostic challenge and need more specific and dedicated paper to discuss this important disease.

We add a brief sentences in the text (introduction section):

“Specific conditions such as pediatric myocarditis, Kawasaki disease, rheumatic carditis, Chagas disease, myasthenia gravis, thyrotoxicosis, polymyositis and other rare infectious forms (parasites, protozoa, fungi), deserve separate discussion and are not addressed in this document”

There are multiple issues with respect to the use of the English language, for example see line 17, 22, 270, 338, 356, 361, 362, 372, 375, etc. It would be most important that the paper is reviewed carefully by a native English speaker prior to resubmitting the paper to the Journal. You may also need to comment as to whether the figures that you have provided are ones that you have created yourself or whether they come from previous articles. If the latter it would be important that you obtain copyright permission for them to be published in another Journal.

As rightly suggested, we have the native English-speaking Author that revise the text again to improve it, correcting the passages highlighted by the reviewer. We certify that the figures are all created by ourself and do not come from previous articles or publish elsewhere.